# Length and Energy Dependence of Low-Energy Electron-Induced Strand Breaks in Poly(A) DNA

**DOI:** 10.3390/ijms21010111

**Published:** 2019-12-23

**Authors:** Kenny Ebel, Ilko Bald

**Affiliations:** 1Department 1-Analytical Chemistry and Reference Materials, BAM Federal Institute for Materials Research and Testing, Richard-Willstätter-Str. 11, 12487 Berlin, Germany; keebel@uni-potsdam.de; 2Institute of Chemistry, Physical Chemistry, University of Potsdam, Karl-Liebknecht-Str. 24-25, 14476 Potsdam, Germany

**Keywords:** DNA origami, DNA radiation damage, DNA strand breaks, low-energy electrons, sequence dependence

## Abstract

The DNA in living cells can be effectively damaged by high-energy radiation, which can lead to cell death. Through the ionization of water molecules, highly reactive secondary species such as low-energy electrons (LEEs) with the most probable energy around 10 eV are generated, which are able to induce DNA strand breaks via dissociative electron attachment. Absolute DNA strand break cross sections of specific DNA sequences can be efficiently determined using DNA origami nanostructures as platforms exposing the target sequences towards LEEs. In this paper, we systematically study the effect of the oligonucleotide length on the strand break cross section at various irradiation energies. The present work focuses on poly-adenine sequences (d(A_4_), d(A_8_), d(A_12_), d(A_16_), and d(A_20_)) irradiated with 5.0, 7.0, 8.4, and 10 eV electrons. Independent of the DNA length, the strand break cross section shows a maximum around 7.0 eV electron energy for all investigated oligonucleotides confirming that strand breakage occurs through the initial formation of negative ion resonances. When going from d(A_4_) to d(A_16_), the strand break cross section increases with oligonucleotide length, but only at 7.0 and 8.4 eV, i.e., close to the maximum of the negative ion resonance, the increase in the strand break cross section with the length is similar to the increase of an estimated geometrical cross section. For d(A_20_), a markedly lower DNA strand break cross section is observed for all electron energies, which is tentatively ascribed to a conformational change of the dA_20_ sequence. The results indicate that, although there is a general length dependence of strand break cross sections, individual nucleotides do not contribute independently of the absolute strand break cross section of the whole DNA strand. The absolute quantification of sequence specific strand breaks will help develop a more accurate molecular level understanding of radiation induced DNA damage, which can then be used for optimized risk estimates in cancer radiation therapy.

## 1. Introduction

Ionizing radiation is used in cancer radiation therapy to damage the DNA of tumors and to reduce the tumor tissue. To optimize the irradiation modality in cancer radiation therapy, a simulation of the dose distribution in the patient is required, which must be based on the fundamental physical processes involved in the interaction of high-energy quanta with the biological materials. Consequently, there is a need of accurate quantification of DNA radiation damage in the form of absolute cross sections for certain processes, such as radiation-induced DNA strand breaks (SBs). During the last two decades, it was shown that the radiation damage induced by high-energy primary radiation is mostly due to the reaction of secondary species generated along the ionization track of water molecules. Secondary species such as hydroxyl radicals or low-energy electrons (LEEs) belong to the most important intermediates of water radiolysis [1]. Most of the LEEs possess an energy below 20 eV with the most probable energy close to 10 eV [2]. At such low electron energies, resonant formation of DNA strand breaks is observed. In an aqueous environment, the LEEs are further thermalized until they reach an energy close to zero eV (quasi-free electrons) and are solvated by water [3].

An LEE can directly attach to DNA by occupying a formerly unoccupied molecular orbital via dissociative electron attachment (DEA). The DEA process is an effective dissociative reaction pathway, where a transient negative ion (TNI) is formed via a resonant Franck-Condon transition [4,5]. The short living TNI can decay by dissociation of a chemical bond. This bond cleavage results in a negative ion and one or more neutral fragments, which is exemplified in Equation (1) for a polyadenine nucleotide (dA_m_).
(1)$$dA_{m}+e^{-}\to dA_{m}^{\#-}\to dA_{m-n}^{-}+dA_{n}$$

The index “#” represents the transient anionic state. In a macromolecule such as DNA, it is also possible that the bond cleavage does not lead to separate fragments.

However, if the dissociation takes place within the phosphodiester bond of the DNA backbone, the bond cleavage represents a DNA SB. Single SBs (SSBs) can be often repaired by proteins [6], while double SBs (DSBs) typically result in apoptosis (cell death) [7]. A common model to study DNA SSBs and DSBs is plasmid DNA. Boudaïffa et al. published one of the first studies of electron-induced DNA strand breaks in the energy range of 3 to 20 eV [2]. The supercoiled plasmid DNA, which consists of several thousand base pairs, can be deposited on a surface and is exposed to radiation. The fragmented DNA is separated by agarose gel electrophoresis (AGE) into its different morphologies (circular, linear, and short fragments) representing different states of damage (SSBs, DSBs, and multiple strand breaks). This analytical method can detect SSBs and DSBs with very high sensitivity. However, it is unclear how the secondary structure affects the damage and which specific DNA sequence is damaged. Traditional chemical analysis tools for oligonucleotides such as HPLC (high performance liquid chromatography) are able to identify the specific bonds that are broken, but have a limited sensitivity and only very short oligonucleotides up to 4 nucleotides (nt), which can be efficiently analyzed [8]. Hence, to determine effects of the DNA sequence on electron-induced DNA strand break yields, an alternative technique is required. Therefore, we have used the DNA origami technique in the present work (Figure 1a) [9]. Well-defined DNA targets attached to DNA origami platforms can be used to determine absolute cross sections for SBs by atomic force microscopy (AFM) (Figure 1b). The DNA origami technique has several advantages compared to other experimental approaches: (i) It allows for maximum sensitivity and only small amounts of material are required. (ii) Within a single irradiation experiment, two different oligonucleotide sequences can be directly compared under the same experimental conditions. (iii) Absolute DNA strand break cross sections for specific sequences can be determined. (iv) The DNA nanoarrays can be modified not only with single stranded DNA (ssDNA), but also with double stranded DNA (dsDNA) or higher-order DNA structures [10]. (v) Application to other radiation sources such as UV radiation is possible [11]. These properties give us the opportunity to study interactions of different well-defined DNA molecules with LEEs of varying energies.

Recently, it was shown that low-energy electron-induced DNA strand breaks depend, to some degree, on the specific DNA sequence and length [11]. Four homo-oligonucleotides (d(A_12_), d(G_12_), d(C_12_), and d(T_12_)) have been compared and the SB cross section upon irradiation with 8.8 eV electrons was the highest for the dA_12_ sequence, even though the differences were small. In another experiment, guanine (G)-rich telomer sequences (5′-d(GGG ATT)_2_) were placed on the DNA origami platforms and exposed to low-energy electrons (8.8 eV) [10]. A higher sensitivity towards LEEs for longer DNA sequences (going from 5′-d(GGG ATT)_2_ over 5′-d(GGG ATT)_3_ to 5′-d(GGG ATT)_4_) was observed. This effect can either result from the increased number of nucleobases or the increased amount of G bases in strands of identical length. The latter was studied in previous experiments, where the authors showed an increasing sensitivity toward 1 eV electrons with higher G content [12]. In the present work, we extend these studies to systematically investigate the effect of the oligonucleotide length on the SB cross section upon LEE irradiation. We quantify and compare DNA strand breaks for various lengths of polyadenine ssDNA (4, 8, 12, 16, and 20 nucleotides). The DNA target sequences are irradiated with electron energies of 5.0, 7.0, 8.4, and 10 eV, which covers the energy range of secondary electrons produced in water radiolysis and which are close to typical energies at which SSBs have been observed before [13]. This yields information about resonant formation of DNA strand breaks for specific ssDNA and a presumed additivity of SB cross sections.

## 2. Results

Triangular DNA origami nanostructures, designed by P. W. Rothemund [14], were used as platforms for our target DNA sequences. Each DNA origami can carry six biotinylated target sequences, including three that have the same sequence, because their positions are indistinguishable in AFM images. The DNA origami structures are adsorbed on silicon substrates and, after irradiation with LEEs in a high vacuum chamber, the intact DNA target sequences on the DNA origami platforms are visualized for AFM images by incubation with streptavidin (SAv, see Figure 1b). We chose five different mono nucleobase DNA sequences with four, eight, 12, 16, and 20 adenine units: 5′-d(A_4_), 5′-d(A_8_), 5′-d(A_12_), 5′-d(A_16_), and 5′-d(A_20_). Figure 2 shows examples of exposure response curves showing the linear dependence of the relative number of strand breaks (N_SB_) on the fluence. From the slope of the linear fits, we determined the absolute SB cross sections for the respective DNA sequence and electron energy (see Methods section for a detailed explanation). Table 1 summarizes all experimentally determined DNA strand break cross sections. In the following, we will first discuss the energy dependence of SB cross sections, and then the dependence on the oligonucleotide length.

### 2.1. Electron Energy Dependence of Strand Break Cross Sections

Figure 3 shows the SB cross sections as a function of electron energy between 5.0 and 10.0 eV. All determined DNA strand break cross sections *σ*_SB_ are in the regime of 10^−15^ cm^2^ with the highest cross sections for d(A_16_) at all electron energies. Below the ionization threshold, DNA bases can either be damaged by pulsed lasers (~4.6 eV) [15,16], Klicken oder tippen Sie hier, um Text einzugeben. which results in oxidative damage to the DNA, or at specific electron energies through anion resonances, i.e., TNI states, which can either decay by autodetachment (AD) of the extra electron or dissociation (DEA). The longer the lifetime of the generated TNI is, the more likely it is that dissociation dominates over AD [4]. Electrons with certain energy can attach e.g., to base π* orbitals, which can serve as antennas for low-energy electrons. At low electron energies, so-called shape resonances are formed, which can result in a transfer of the extra electron to the DNA backbone and eventually in cleavage of sugar-phosphate *σ*(C-O) bonds [17]. The strand break cross sections for all sequences investigated in this section exhibit a broad resonant structure peaking at 7.0–8.4 eV. In this regime, all DNA subunits give rise to various resonant fragmentation pathways. At these energies, the reactions are mediated by core excited resonances in which the electron attachment is accompanied by an electronic excitation. The lowest-energy electronic excitation in DNA represents a π-π* transition at about 4.7 eV, which is then the low-energy threshold for a core excited resonance in DNA [5]. In the present experiment, the lowest cross sections are observed at 5.0 eV. A similar and strong electron energy dependent signature of SB yield was shown in several studies using plasmid DNA [2,13,18]. In contrast to the single-stranded oligonucleotides used in our experiment, plasmid DNA is a double stranded DNA with several thousand base pairs, a complex secondary structure, and is prepared as a several nm thick film. In all these studies, the authors observed a broad resonance maximum between 7.0 and 13 eV for SSBs, which is comparable to our experiment. At higher electron energies, DNA strand break yields are slightly decreasing and have a minimum at 14.0 to 15.0 eV. The same pronounced peak signatures for different forms of damages below 15 eV have been previously observed from condensed phase DNA bases [19] and from single-stranded oligonucleotides adsorbed on gold surfaces [20] and immobilized on DNA origamis [21].

The vertical ionization energy (IE) of adenine is 8.4 eV [22], i.e., upon irradiation with LEEs at 8.4 eV or higher. Strand breakage could also result from initial ionization of the DNA strand. In other experiments, using vacuum ultraviolet (VUV) radiation at 8.4 eV and the same target sequences as in the present study, similar tendencies of the SB cross sections have been observed: d(A_4_) ≈ d(A_8_) < d(A_12_) < d(A_16_) ≈ d(A_20_). VUV induced strand breaks are initiated by dissociative photo ionization and excitation [11,23]. However, irradiation with LEEs yields DNA strand break cross sections, which are one order of magnitude higher than for VUV radiation demonstrating the high efficiency of DEA compared to dissociative photo excitation and ionization. Therefore, it is unlikely that, in the present experiments using LEEs up to 10 eV, other processes than DEA contribute.

### 2.2. Oligonucleotide Length Dependence of Strand Break Cross Sections

It is expected that low-energy electron-induced DNA strand breaks strongly depend on the specific oligonucleotide length. Under the high-vacuum conditions during LEE irradiation, the DNA is likely to be present as A-DNA. Assuming a length of 0.24 nm per base pair (bp) and a width of 1.15 nm of linear single stranded A-DNA [24], we can estimate the geometrical cross section for each investigated sequence and compare this with the SB cross sections obtained in our experiments (Figure 4). No matter which exact molecular model is used, we are generally expecting an increase of the DNA strand break cross section with the oligonucleotide length because of the higher probability of attaching an electron and a greater number of possible bond breakages. Upon electron attachment to DNA, several different reaction pathways are possible, and strand breakage is only one possibility. Consequently, the estimated geometrical cross section is about one order of magnitude higher than the experimentally determined SB cross sections. Figure 4 shows a plot of the experimentally determined DNA strand break cross sections as a function of the number of nucleotides. The DNA strand break cross sections *σ*_SB_ at 5.0 and 10.0 eV vary with the oligonucleotide length, according to the following row: d(A_4_) ≈ d(A_8_) ≈ d(A_20_) < d(A_12_) < d(A_16_). Within the shortest DNA sequences with four and eight nucleotides, we observe only a moderate change of the SB cross section with the oligonucleotide length with a slight increase from eight over 12 to 16 nucleotides. At electron energies corresponding to the maximum of negative ion resonances (7.0 and 8.4 eV), we determined a stronger increase of the DNA strand break cross section *σ*_SB_ with the oligonucleotide length, according to the following tendency: d(A_4_) < d(A_8_) ≈ d(A_20_) < d(A_12_) < d(A_16_), with an initial slope similar to the geometrical cross section. Nevertheless, the SB cross sections per nucleotide (Table 2) vary strongly for each sequence and irradiation energy from 0.43 to 11.98 ∙ 10^−16^ cm^2^, which indicates that they are not additive under the assumption that each nucleotide is equivalent.

Comparing the DNA sequences d(A_16_) and d(A_20_), the cross sections for SBs do not continue the increasing trend, but decrease significantly for d(A_20_) at all electron energies. Tentatively, we assume that this can be ascribed to a conformational change occurring in the longer oligonucleotides. It was shown previously that A-containing DNA and RNA can form A-duplexes in the presence of ammonium ions and at nearly a neutral pH [25]. Figure 5a shows a scheme of a symmetric base pair formed by two A nucleotides representing the basic binding motif for a parallel double-stranded helix. The two hydrogens of the amino group of one A base form hydrogen bonds with the second A base and the oxygen of the phosphate in the DNA backbone. In total, the A-A base pairing is mediated by four hydrogen bonds supported by the stabilization through the present cations. For the present experiment, we can assume that, as soon as the DNA reaches a certain length, the Coulomb repulsion of the DNA backbone becomes too weak to further establish a linear structure. Due to the presence of 150 mM Mg^2+^ and a pH of 8.0 in the present experiment, a conformational change of d(A_20_) appears feasible. Based on this assumption, the single-stranded DNA would have to form a DNA hairpin with at least three to four nucleotides in the loop. An ssDNA that is 20 nucleotides long and a DNA hairpin with eight A-A base pairs could be formed with a melting point T_m_ near room temperature [26]. When going to a lower number of nucleotides, T_m_ is too low to be stable in this conformation. Figure 5b illustrates the conformational change if the DNA reaches a certain length. Additionally, for the folded d(A_20_) hairpin, one can assume in the first approximation that the geometrical cross section increases compared to d(A_16_). The drop of the SB cross section must then be explained by an increase of the stability of the DNA hairpin due to the additional hydrogen bonds. If an SSB process in the coiled ssDNA takes place, the cleaved part and the biotin label could stay attached. Subsequent AFM analysis is then not able to detect this SSB and the result is a smaller DNA strand break cross section (Figure 5b). The length dependence of the SB cross section upon LEE irradiation was already studied by Rackwitz et al., who used a telomer sequence with two (5′-d(GGG ATT)_2_), three (5′-d(GGG ATT)_3_), and four repeat units (5′-d(GGG ATT)_4_) and used 8.8 eV electron energy. Very generally, a length dependence with increased sensitivity for longer oligonucleotides was found. Due to a change in the secondary structure in the telomer sequence, the DNA strand break cross sections do not follow a linear function. In the presence of cations, telomer sequences are known to fold into different intermolecular G-quadruplexes. It can be expected that, for shorter sequences (5′-d(GGG ATT)_2_ and 5′-d(GGG ATT)_3_), the probability of folded structures is lower than for longer sequences like 5′-(GGG ATT)_4_.

So far, we have discussed the relative changes of SB cross sections for different oligonucleotide lengths. In the following, we will discuss the magnitude of determined absolute cross section values. Panajotovic et al. determined effective cross sections for production of SSBs in supercoiled plasmid DNA upon irradiation with LEEs at 0.1 to 4.7 eV [27]. The authors determined effective SSB cross sections per nucleotide *σ*_N_ in the order of 10^−18^ cm^2^/nucleotide at 1 and ~2 eV electron energy. A similar value was obtained afterward for the absolute cross section for loss of supercoiled DNA upon irradiation with 10 eV electrons (3.0 × 10^−14^ cm^2^ for the whole plasmid corresponding to 4.8 × 10^−18^ cm^2^/nucleotide) [28]. In these studies, it is assumed that every nucleotide has the same DNA strand break cross section and that the total SSB cross section depends linearly on the length of the plasmid. In the present experiments, we use DNA strands with 4 to 20 nucleotides, which results in *σ*_N_ values in the order of 10^−16^ to 10^−17^ cm^2^/nucleotide, which is one to three orders of magnitudes higher than the previous results obtained using plasmid DNA. While the present results indicate that a linear length dependence of SB cross sections can be assumed at the maximum of the negative ion resonances and within a certain length regime, the SB cross sections per nucleotide still vary strongly, which indicates that a simple additivity of SB cross sections per nucleotide is not valid (Table 2). A similar observation was made recently for absolute vibrational excitation cross sections. The cross section values of thymidine do not correspond to the sum of the cross sections of its constituents [29]. To further elucidate the discrepancy in the SB cross sections, further experiments are required using e.g., double-stranded DNA, which has higher conformational stability over a longer range of strand lengths. Furthermore, when comparing the cross section values obtained from the different DNA systems, the potentially different surrounding cations (concentrations and species) and solvating water molecules need to be considered.

## 3. Materials and Methods

Triangular DNA origami has been used because it is one of the most structurally sTable 2D DNA origami, which does not tend to aggregate and adsorbs as flat monomers on surfaces. Six of the original staple strands are extended with our target sequences. For the DNA origami assembly, 5 µL of 100 nM viral scaffold strand M13mp18, 202 short ssDNA (6.7 µL each) are mixed with 10 µL of 10 × TAE-buffer (with 125 mM MgCl_2_), 9 µL of 100 nM modified target sequences, and 46 µL deionized water. During an annealing process, the DNA nanostructures is heated up to 80 °C and cooled down stepwise over 2 h to 6 °C. The solution of self-assembled triangular-shaped DNA origami is filtered three times with 1 × TAE-buffer containing 15 mM MgCl_2_ for 6 min at 4629 g. Each DNA origami substrate can carry two different (black and green in Figure 1) biotinylated target sequences at fixed positions. The assembled DNA nanostructures are immobilized on 8 × 8 mm^2^ air plasma cleaned silicon substrates. The sample was cleaned with 1 mL of 1:1 deionized water/ethanol and dried for one hour in absolute ethanol to get rid of the remaining water. Samples are then transferred to a high vacuum chamber made of stainless steel, which is connected to a turbo pump (TwisTorr 304, Agilent Technologies, Californien, USA) that reaches a pressure of about 10^−8^ mbar. An octagonal sample holder can be rotated and translated to place the samples into the electron beam. A flood gun FS100 generates the LEEs in the range of 5.0 to 500 eV. The electron beam is focused by three electrostatic lenses and illuminates an irradiation area *Z* of 0.78 cm^2^. The electron current on the sample *I*_S_ is measured by a picoamperemeter (6485E by Keithley). Si/SiO_2_ is getting charged by the electron beam, which results in a repulsive potential and reduces the electron energy. Therefore, the current onset was set to be the zero-point of the electron energy scale. The fluence F is the number of electrons n_e_ per irradiated surface area *Z* (Equation (2)).
(2)$$F\left[cm^{-2}\right]=\frac{n_{e}}{Z\left[cm^{2}\right]}$$

The measured current I_S_ and the irradiation time t divided by the elementary charge *e* gives the number of electrons n*_e_* arriving at the sample (Equation (3)).
(3)$$n_{e}=\frac{I\left[nA\right]*t\left[s\right]}{e}$$

Prior to the irradiation experiments, a Faraday cup was used to determine the ratio between the current measured by the Faraday cup (*I*_FC_) and on the sample (*I*_S_). In this way, the absolute number of electrons arriving at the sample can be determined. In a single irradiation experiment, the strand break cross section of at least two different sequences on one DNA origami is obtained. To prove repeatability, a fluence dependency needs to be recorded for at least two different sample series under identical conditions. There are six irradiated samples and two non-irradiated control samples in one sample series [30]. After irradiation, the remaining intact oligonucleotides are visualized by atomic force microscopy (AFM) with streptavidin (SAv), which binds specifically to biotin (Bt). The streptavidin-biotin-bond is one of the strongest known non-covalent interactions in nature, which exhibits a dissociation constant K_d_ of 10^−14^ M–10^−15^ M [31]. The samples were incubated for 2 min with a 50 nM Sav solution in 1 × TAE-buffer with 15 mM MgCl_2_. An AFM was used in the soft tapping mode to not destroy the DNA origamis. Figure 1b shows AFM images of DNA origami nanostructures with a different state of LEE-induced damage. The (0/0) DNA origami represents a non-irradiated control sample where all six target positions are intact. The first number in brackets represents the amount of intact DNA strands in the middle position while the second number represents the side position. The two different positions (middle and side position) on each side of the triangle can be clearly distinguished in AFM images. For this reason, two different DNA strands can be studied under the same irradiation conditions. Each sample is analyzed manually by counting all intact DNA strands. This gives us results for the relative number of strand breaks (*N*_SB_) by comparing the intact DNA strands to a non-irradiated control sample. *N*_SB_ is calculated by the ratio of the number of intact oligonucleotides before (*N*_0_) and after irradiation (*N*_t_) for a specific irradiation time t (Equation (4)) [32].
(4)$$N_{\mathrm{SB}}\;=\;1-\frac{N_{t}}{N_{0}}$$

By recording the fluence dependence of *N*_SB_, the cross section for DNA strand breaks *σ*_SB_ can be determined from the slope of the linear fit in the low-fluence regime.
(5)$$\sigma_{\mathrm{SB}}=\;\frac{N_{SB}}{F\left[cm^{-2}\right]}$$

Figure 2 shows examples of exposure-response curves to determine the absolute DNA strand break cross section. The chosen fluence is small enough that no saturation of *N*_SB_ is reached, which is the result of crosslinked DNA on DNA origami [9]. The technique is highly reproducible, which yields uncertainties of typically 10% to 20%.

## 4. Conclusions

We presented absolute cross sections for SBs in polyA sequences of different lengths (with 4, 8, 12, 16, and 20 nucleotides) upon LEE irradiation with 5.0, 7.0, 8.4, and 10.0 eV. The SB cross sections were found to be in the range of 10^−15^ cm^2^ with the highest cross sections for d(A_16_) at all electron energies. A broad resonant structure of SB cross section is found with a maximum between 7.0 and 8.4 eV for all polyA sequences.

The data allows for a systematic analysis of the length dependency of SB cross sections. Because of the higher probability of attaching an electron and a greater number of possible bond breakages in longer oligonucleotides, an increase of the DNA strand break cross section is expected with the length and is basically confirmed by the experiment. However, the length dependency of the SB cross section is found to be dependent on the electron energy. At electron energies of 7.0 and 8.4 eV, a stronger increase of the DNA strand break cross section *σ*_SB_ compared to 5.0 and 10.0 eV is obtained, with an initial slope similar to the geometrical cross section. Nevertheless, the SB cross section per nucleotide varies strongly with oligonucleotide length and electron energy, which indicates that the individual nucleotides do not contribute independently to the SB cross section. Consequently, the normalization of the SB cross section to the number of nucleotides in large macromolecules such as plasmid DNA might be misleading, and, in this case, the SB cross section for a whole macromolecule is more meaningful.

Furthermore, we have found that the cross sections for SBs depend strongly on the DNA conformation, i.e., the SB cross section of d(A_20_) is significantly smaller than for d(A_16_). We assume that, as soon as the DNA reaches a certain length, then the Coulomb repulsion of the DNA backbone is too weak to further establish a linear structure and a conformational change of d(A_20_) is possible. The DNA starts to coil, which creates stabilizing hydrogen bonds. Subsequent AFM analysis is then unable to detect an SB in the coiled ssDNA because the cleaved part and the biotin label could stay attached to the DNA target sequence. The result is a smaller DNA strand break cross section.

In future experiments, this phenomenon could be studied in more detail using dsDNA, which is conformationally more stable than ssDNA.

## Figures and Tables

**Figure 1 ijms-21-00111-f001:**
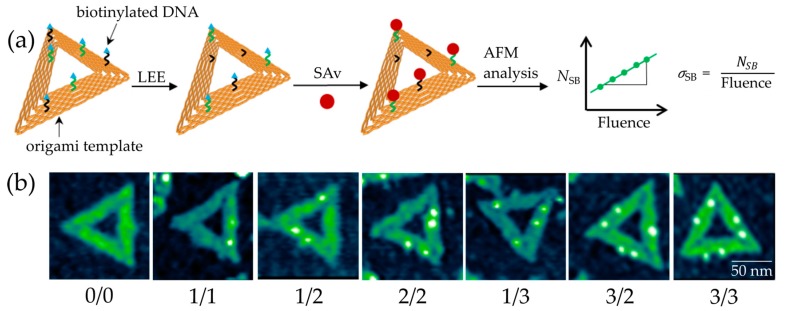
(**a**) Scheme of the DNA origami structure and the experimental procedure to determine the absolute cross section of DNA strand breakage. Each DNA origami can carry two different (black and green) biotinylated target sequences. Incubation with streptavidin (SAv) marks the intact DNA sequences. (**b**) AFM images of DNA origami nanostructures irradiated at 7 eV electron energy. Each bright spot indicates an intact DNA target sequence.

**Figure 2 ijms-21-00111-f002:**
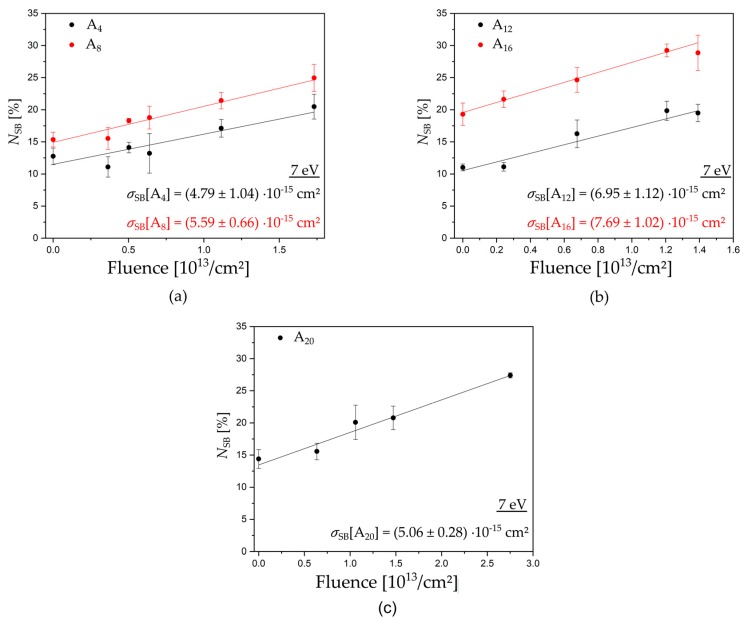
Examples of exposure response curves to determine the absolute DNA strand break cross section for ssDNA. Irradiation was performed at 7 eV electron energy for the five oligonucleotide sequences (**a**) d(A_4_) black and d(A_8_) red, (**b**) d(A_12_) black and d(A_16_) red, and (**c**) d(A_20_) black. From the linear fit of the fluence and the dependence of strand breakage, the absolute strand break cross section can be obtained.

**Figure 3 ijms-21-00111-f003:**
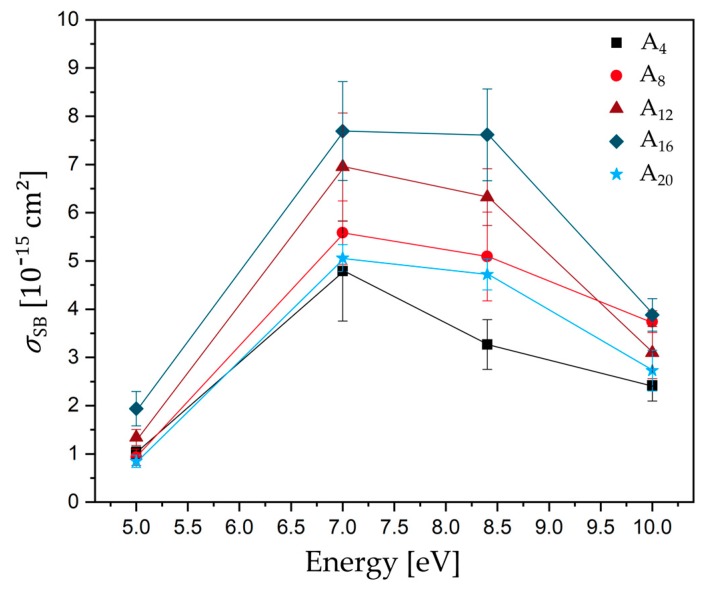
Plot of absolute DNA strand break cross section for different single stranded DNA sequences (d(A_4_) black-square, d(A_8_) light red-circle, d(A_12_) dark red-triangle, d(A_16_) dark blue-diamond, and d(A_20_) light blue-star) as a function of the electron energy. Solid lines connect the data points to guide the eye.

**Figure 4 ijms-21-00111-f004:**
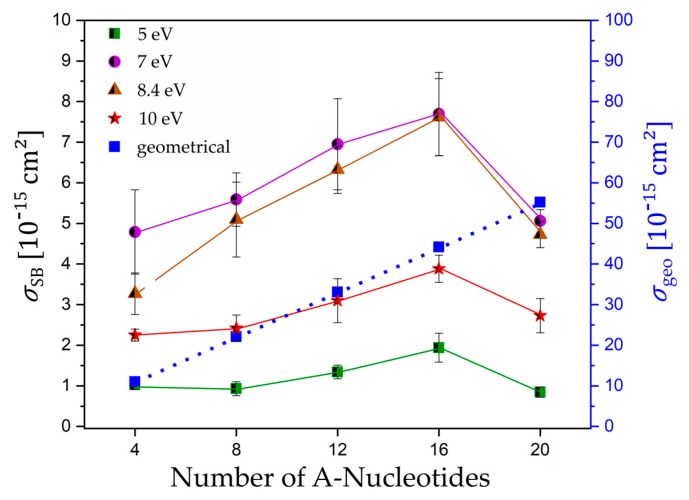
Absolute cross section for SBs (black) and the estimated geometrical cross section (blue) for all DNA sequences 5′-d(A_n_) *n* = 4, 8, 12, 16, 20 irradiated at 5.0, 7.0, 8.4, and 10 eV plotted against the number of nucleotides. Solid and dotted lines connect the data points to guide the eye.

**Figure 5 ijms-21-00111-f005:**
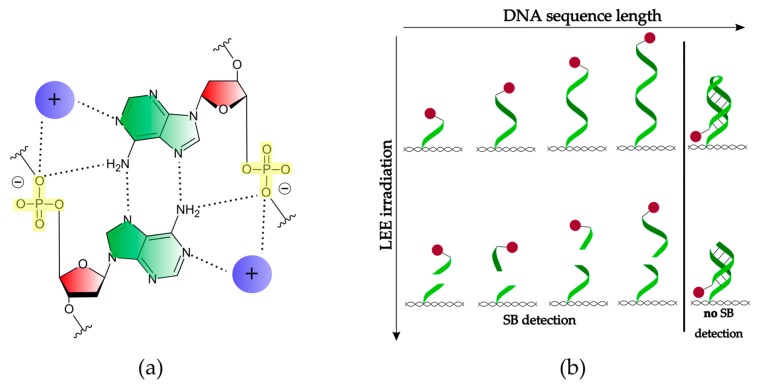
(**a**) Scheme of the A-A (green) base pair attached to the deoxyribose sugar unit (red) that builds the poly(A) duplex in the presence of cations (purple). The phosphate groups (yellow) are driven toward the axis of the helix and form two of the total four hydrogen bonds (dotted lines) [25]. (**b**) Scheme of the conformational change of a single-stranded DNA with increasing length and SB formation. DNA origami is depicted as a pattern in black, the biotin label as a spot in red, and the hydrogen bonding is labeled as a solid line in black.

**Table 1 ijms-21-00111-t001:** Overview of the absolute cross sections for SBs(***σ***_SB_) for the different DNA sequences upon electron irradiation at different energies compared to the geometrical cross section (***σ*_geo_**).

Sequence	ssDNA SB Cross Section *σ*_SB_ [10^−15^ cm^2^]				*σ*_geo_ [10^−15^ cm^2^]
	5 eV	7 eV	8.4 eV	10 eV	/
5′-dA_4_	1.03 ± 0.12	4.79 ± 1.04	3.27 ± 0.51	2.25 ± 0.15	11.04
5′-dA_8_	0.93 ± 0.17	5.59 ± 0.66	5.09 ± 0.92	2.42 ± 0.33	22.08
5′-dA_12_	1.34 ± 0.17	6.95 ± 1.12	6.32 ± 0.59	3.10 ± 0.54	33.12
5′-dA_16_	1.94 ± 0.36	7.69 ± 1.02	7.62 ± 0.95	3.88 ± 0.34	44.16
5′-dA_20_	0.85 ± 0.13	5.06 ± 0.28	4.72 ± 0.32	2.73 ± 0.42	55.20

**Table 2 ijms-21-00111-t002:** Overview of absolute SB cross sections per nucleotide for experimental data *σ*_N_ and geometrical cross section *σ*_N-geo_ calculated from the determined values in Table 1.

Sequence	SB Cross Section per Nucleotide *σ*_N_ (10^−16^ cm^2^)				*σ*_N-geo_ (10^−16^ cm^2^)
	5 eV	7 eV	8.4 eV	10 eV	/
5′-dA_4_	2.58	11.98	8.18	5.63	27.60
5′-dA_8_	1.16	6.99	6.36	3.03	27.60
5′-dA_12_	1.12	5.79	5.27	2.58	27.60
5′-dA_16_	1.21	4.81	4.76	2.43	27.60
5′-dA_20_	0.43	2.53	2.36	1.37	27.60

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
