# Peer review of "Length and Energy Dependence of Low-Energy Electron-Induced Strand Breaks in Poly(A) DNA"

_ijms, 2019, doi:10.3390/ijms21010111_

Round 1

Reviewer 1 Report

This manuscript is a timely and very interesting manuscript on the effect of low-energy electron exposure on adenine-containing DNA and how this depends on the electron’s energy and the DNA’s length. The manuscript is well written and structured and the methods used are appropriate and well described. I recommend publication with a few minor comments to address a couple points I found could be slightly improved: 

In line 62 it is suggested that “single SBs (SSBs) can be often repaired by proteins, while double SBs (DSBs) typically result in apoptosis (cell death)”. I believe one or two references supporting this statement would be helpful. In line 76 onwards the DNA origami technique is discussed over its several advantages with respect to other experimental approaches. I believe it would be positive for completeness to briefly mention those other techniques that might not be as good, and if there are aspects in which they could be preferable, particularly in terms of studying the larger sequences (A20) which are shown to be tricky within the technique used.  In line 101-104, both sentences start with “In the present work”. A better way to phrase it could be found. In Table 1 caption it would help to specify that cross sections are labeled as sigma, i.e. one could add (sigma SB) and (sigma geo) after they are mentioned in Table 1 caption to help the reader identify these.  In line 147, the possibility of damaging DNA with sources below the ionisation threshold are briefly discussed. Here it is worth mentioning the recent work of Markovitsi and co-workers [Faraday Discuss 207, 181 (2018); J Phys Chem B 123, 4950 (2019)]: they show how 266nm (~4.6 eV) laser pulses are able to significantly ionise bases in double strands.   In lines 200-203 Figure 4 is briefly discussed. In Figure 4 it can be seen how there are two very different trends: those for samples at 7 and 8.4 eV, and those at 5 and 10 eV. Whereas the former are discussed, the latter are not mentioned. It is previously discussed in the text how 7 and 8.4 eV are resonant with the maximum of negative ion resonances, but it is still interesting to see how 5 and 10 eV yield effectively the same result despite one being twice as energetic as the other, and is something the authors could perhaps further elaborate on. 

Author Response

In line 62 it is suggested that “single SBs (SSBs) can be often repaired by proteins, while double SBs (DSBs) typically result in apoptosis (cell death)”. I believe one or two references supporting this statement would be helpful.

Authors reply: The following references are added to the article.

Caldecott, K. W. DNA single-strand break repair. Experimental cell research 2014, 329 (1), 2–8. Pandey, M.; Raghavan, S. DNA double-strand break repair in mammals. J Radiat Cancer Res 2017, 8 (2), 93.

In line 76 onwards the DNA origami technique is discussed over its several advantages with respect to other experimental approaches. I believe it would be positive for completeness to briefly mention those other techniques that might not be as good, and if there are aspects in which they could be preferable, particularly in terms of studying the larger sequences (A20) which are shown to be tricky within the technique used.  

Authors reply 66-73: To complete the overview over the available techniques, we added positive and negative aspects for AGE and HPLC.

“The fragmented DNA is separated by agarose gel electrophoresis (AGE) into its different morphologies (circular, linear and short fragments) representing different states of damage (SSBs, DSBs and multiple strand breaks). This analytical method can detect SSBs and DSBs with very high sensitivity, however, it is unclear how the secondary structure affects the damage and which specific DNA sequence is damaged. Traditional chemical analysis tools for oligonucleotides such as HPLC are able to identify the specific bonds which are broken, but have a limited sensitivity and only very short oligonucleotides up to 4 nucleotides (nt) can be efficiently analyzed [8]“

In line 101-104, both sentences start with “In the present work”. A better way to phrase it could be found.

Authors reply 105-108: We agree with the reviewers comment and changed the phrases.

“In the present work we extend these studies to investigate systematically the effect of the oligonucleotide length on the SB cross section upon LEE irradiation. We quantify and compare DNA strand breaks for various lengths of polyadenine ssDNA (4, 8, 12, 16 and 20 nucleotides). The DNA target sequences are irradiated with electron energies of 5.0, 7.0, 8.4 and 10 eV, which covers the energy range of secondary electrons produced in water radiolysis and which are close to typical energies at which SSBs have been observed before [13].“

In Table 1 caption it would help to specify that cross sections are labeled as sigma, i.e. one could add (sigma SB) and (sigma geo) after they are mentioned in Table 1 caption to help the reader identify these.  

Authors reply: The caption of table 1 and 2 was changed as follows:

“Table 1. Overview of the absolute cross sections for SBs (σSB) for the different DNA sequences upon electron irradiation at different energies compared to the geometrical cross section (σgeo).

Table 2. Overview of absolute SB cross sections per nucleotide for experimental data σN and geometrical cross section σN-geo calculated from the determined values in table 1.”

In line 147, the possibility of damaging DNA with sources below the ionisation threshold are briefly discussed. Here it is worth mentioning the recent work of Markovitsi and co-workers [Faraday Discuss 207, 181 (2018); J Phys Chem B 123, 4950 (2019)]: they show how 266nm (~4.6 eV) laser pulses are able to significantly ionise bases in double strands.  

Authors reply 135-139: The following references are added to the article. Additionally, minor changes due to the newly added references have been made.

Banyasz, A.; Balanikas, E.; Martinez-Fernandez, L.; Baldacchino, G.; Douki, T.; Improta, R.; Markovitsi, D. Radicals Generated in Tetramolecular Guanine Quadruplexes by Photoionization: Spectral and Dynamical Features. The journal of physical chemistry. B 2019, 123 (23), 4950–4957. Banyasz, A.; Ketola, T.; Martínez-Fernández, L.; Improta, R.; Markovitsi, D. Adenine radicals generated in alternating AT duplexes by direct absorption of low-energy UV radiation. Faraday Discuss. 2018, 207 (30), 181–197.

“Below the ionization threshold DNA bases can either be damaged by pulsed lasers (~4.6 eV) [15], [16] resulting in oxidative damage to the DNA or at specific electron energies through anion resonances, i.e. TNI states which can either decay by autodetachment (AD) of the extra electron or dissociation (DEA).“

 In lines 200-203 Figure 4 is briefly discussed. In Figure 4 it can be seen how there are two very different trends: those for samples at 7 and 8.4 eV, and those at 5 and 10 eV. Whereas the former are discussed, the latter are not mentioned. It is previously discussed in the text how 7 and 8.4 eV are resonant with the maximum of negative ion resonances, but it is still interesting to see how 5 and 10 eV yield effectively the same result despite one being twice as energetic as the other, and is something the authors could perhaps further elaborate on.

Authors reply 184-188: The results described in this article reflect a resonant character at 7 and 8.4 eV. In addition, a low strand break yield at 5 and 10 eV could be determined, which is due to the limited formation of ions. This fact is described in detail in lines 198-202.

“The DNA strand break cross sections σSB at 5.0 and 10.0 eV vary with the oligonucleotide length according to the following row: d(A4) ≈ d(A8) ≈ d(A20) < d(A12) < d(A16). Within the shortest DNA sequences with four and eight nucleotides, we observe only a moderate change of the SB cross section with the oligonucleotide length with a slight increase from eight over twelve to sixteen nucleotides.”

The fact that the strand break cross sections are of similar value for 5 eV and 10 eV indicates that the damage occurs via negative ion resonances as is described in section 2.1.

Reviewer 2 Report

This manuscript present data that may be of a limited interest of IJMS readers as the journal strongly emphasizes research in molecular biology and molecular medicine and this manuscript presents a typical physico-chemical study. DNA in this research is anhydrous in it’s a-like shape, but it is known that low and moderate energy ionizing radiation induce biologic DNA damage-related effects mainly through water radiolysis. Moreover, the authors use a highly artificial DNA – poly(A) a duplex with no a direct relationship to biological conditions. By the way: the authors should use “Poly(A) Duplex” in the title instead of “Adenine Containing DNA”. I do not question the scientific value of this manuscript, the more I think that it present a deep insight into mechanism of DNA breaks induction by ionizing radiation and other factors, but it should be submitted to a journal specializing in physical chemistry. I also wonder on the novelty of this manuscript as compared with a previous manuscript of these authors (PMID: 30719805) in the context of general message, not technical and methodological details.

Author Response

This manuscript present data that may be of a limited interest of IJMS readers as the journal strongly emphasizes research in molecular biology and molecular medicine and this manuscript presents a typical physico-chemical study. DNA in this research is anhydrous in it’s a-like shape, but it is known that low and moderate energy ionizing radiation induce biologic DNA damage-related effects mainly through water radiolysis. Moreover, the authors use a highly artificial DNA – poly(A) a duplex with no a direct relationship to biological conditions. By the way: the authors should use “Poly(A) Duplex” in the title instead of “Adenine Containing DNA”.

Authors reply 2-4: According to the suggestion by the reviewer we changed the article title.

“Length and Energy Dependence of Low-Energy Electron-Induced Strand Breaks in Poly(A) DNA”

I do not question the scientific value of this manuscript, the more I think that it present a deep insight into mechanism of DNA breaks induction by ionizing radiation and other factors, but it should be submitted to a journal specializing in physical chemistry.

Authors reply: This article represents an invited contribution to the Special Issue "Radiation-Induced Damage to DNA", and is therefore considered to be appropriate.

 I also wonder on the novelty of this manuscript as compared with a previous manuscript of these authors (PMID: 30719805) in the context of general message, not technical and methodological details.

Authors reply: The paper mentioned by the reviewer deals with the interaction of vacuum UV and LEE radiation with DNA. The published results show DNA oligonucleotides of different DNA nucleobases (A, C, G, T) with the same DNA length (12 nucleotides). No significant sequence dependence is observed. In addition, the published article shows no energy dependence (VUV 8.44 eV, LEE 8.8 eV) of the DNA strand breakage, but discusses the differences between low-energy electron and VUV photon induced damage. In the present manuscript, however, the question of the lengths and energy dependence of strand breakage in poly(A) DNA is discussed.

Reviewer 3 Report

In the introduction, the authors mention about "The supercoiled plasmid DNA, which consist of several thousand base pairs can be deposited on a surface and is exposed to radiation. The fragmented DNA is separated by agarose gel electrophoresis (AGE) into its different morphologies representing different states of damage". It is unclear what different states of damage and morphologies are referred to here. 

This work is an extension of the previously done work on homo- and hetero-oligonucleotide DNA sequences of varying length and is done with poly adenine sequences. It would be more informative to include varying stretches other nucleobases in the study.

It is unclear why in Figure 2. the exposure response curves for the five oligonucleotide sequences are represented separately in 3 plots.

Overall, the manuscript can be improved with making these changes.

Author Response

In the introduction, the authors mention about "The supercoiled plasmid DNA, which consist of several thousand base pairs can be deposited on a surface and is exposed to radiation. The fragmented DNA is separated by agarose gel electrophoresis (AGE) into its different morphologies representing different states of damage". It is unclear what different states of damage and morphologies are referred to here. 

Authors reply 66-73: To clarify the different states of DNA damage and their morphology, the text has been adapted and further information was added.

“The fragmented DNA is separated by agarose gel electrophoresis (AGE) into its different morphologies (circular, linear and short fragments) representing different states of damage (SSB, DSB and multiple strand breaks MSB). This analytical method can detect SSBs and DSBs with very high sensitivity, however, it is unclear how the secondary structure affects the damage and which specific DNA sequence is damaged. Traditional chemical analysis tools for oligonucleotides such as HPLC are able to identify the specific bonds which are broken, but have a limited sensitivity and only very short oligonucleotides up to 4 nucleotides (nt) can be efficiently analyzed [8]“

This work is an extension of the previously done work on homo- and hetero-oligonucleotide DNA sequences of varying length and is done with poly adenine sequences. It would be more informative to include varying stretches other nucleobases in the study.

Authors reply220-228: In the present article, we investigate the length and energy dependence of DNA strand breakage. In order to be able to draw clear conclusions, we have to focus on the same kind of sequence, that can be easily extended without adding sequence-dependent effects. Therefore, we focused on a simple poly(A) sequence. In earlier studies we focused on the effect of different sequences on the strand break cross section. Basically, the different nucleobases show predominantly the same results, only specific sequences such as the telomeric DNA (ref 10) show pronounced sequence effects.

It is unclear why in Figure 2. the exposure response curves for the five oligonucleotide sequences are represented separately in 3 plots.

Authors reply: Figure 2 shows the exposure-response curves for the DNA sequences A4, A8, A12, A16 and A20 at an electron energy of 7 eV. With the DNA origami technique, two different DNA sequences can be examined under the same experimental conditions. The results from single irradiation experiments are shown in the individual plots of figure 2 (e.g. plot top left: A4 and A8); for this reason, the plots were not merged. In addition, the authors find the combination of all results in a single diagram too confusing.